# Reduced glutathione levels in *Enterococcus faecalis* trigger metabolic and transcriptional compensatory adjustments during iron exposure

Víctor Aliaga-Tobar,[1,2,3] Jorge Torres,[2,3] Sebastián Nelson Mendoza,[4,5] Gabriel Gálvez,[2,3] Jaime Ortega,[2,3] Sebastián Gómez,[2,3] Valentina Parra,[3,6,7] Felipe Arenas,[8] Alejandro Maass,[4,5,9] Anne Siegel,[10] Mauricio González,[5,11] Mauricio Latorre[2,3,4,5,11]

ABSTRACT  *Enterococcus faecalis,* a facultative anaerobic pathogen and common constituent of the gastrointestinal microbiota, must navigate varying iron levels within the host. This study explores its response to iron supplementation in a glutathione-deficient mutant strain (Δ*gsh*). We examined the transcriptomic and metabolic responses of a glutathione synthetase mutant strain (Δ*gsh*) exposed to iron supplementation, integrating these data into a genome-scale metabolic model (GSMM). Our results show that under glutathione deficiency, *E. faecalis* reduces intracellular iron levels and shifts its transcriptional response to prioritize energy production genes. Notably, basal metabolites, including arginine, increase. The GSMM highlights the importance of arginine metabolism, particularly the *arc* operon (anaerobic arginine catabolism), as a presumed compensatory mechanism for glutathione deficiency generated during iron exposure. These findings provide insights into how *E. faecalis* adjusts metal homeostasis and transcriptional/metabolic processes to mitigate the effects of oxidative stress caused by iron.

IMPORTANCE  Iron is essential for bacterial survival, yet its excess can be harmful due to its role in increasing oxidative stress. *Enterococcus faecalis*, a common member of the human gut microbiota, must carefully balance its iron levels to survive in changing environments. Here, we investigate how *E. faecalis* compensates for the reduced availability of glutathione, a key antioxidant, when exposed to high iron concentrations. We discovered that *E. faecalis* lowers its intracellular iron levels when glutathione biosynthesis is disrupted and reprograms its metabolism to prioritize energy production, potentially to fuel stress response mechanisms under iron-induced oxidative conditions. These findings enhance our understanding of bacterial adaptation under oxidative stress and suggest that interfering with arginine metabolic pathways could represent novel strategies to combat *E. faecalis* infections.

KEYWORDS  transcriptome, metabolome, glutathione, iron, *Enterococcus faecalis*

Since the emergence of oxygen on Earth, cells have faced the dual-edged nature of O₂. On one hand, oxygen serves as the terminal electron acceptor in a series of intricate redox reactions, such as oxidative phosphorylation. On the other hand, it is a stable allotropic molecule with an electronic configuration containing two unpaired electrons, capable of generating reactive oxygen species (ROS) (1, 2). These ROS are particularly toxic to cells due to their high reactivity, causing damage to cellular components, including DNA, membrane lipids, and proteins, a process known as oxidative stress, which can ultimately lead to cell death (3). Given the omnipresence of oxidative stress in biological systems, bacteria have evolved sophisticated mechanisms

**Peer Reviewers** Rajib Saha, University of Nebraska-Lincoln, Lincoln, Nebraska, USA; Jessica R. Sheldon, University of Saskatchewan, Saskatoon, Saskatchewan, Canada

Address correspondence to Mauricio Latorre, mauricio.latorre@uoh.cl.

The authors declare no conflict of interest.

See the funding table on p. 18.

to mitigate its harmful effects, particularly in metal-rich environments, which intensify oxidative damage by promoting the formation of ROS. However, the precise role of key antioxidant molecules in modulating global transcriptional responses under such conditions remains underexplored.

In bacteria growing under aerobic conditions, a portion of the cellular damage caused by oxidative stress can be attributed to ROS formed endogenously through reactions between $O_2$ and univalent electron donors, such as metals (4). Among these metals, iron is particularly harmful. Although essential for various metabolic processes due to its redox activity (5), excess iron can become toxic by catalyzing the Fenton and/or Haber-Weiss reactions with hydrogen peroxide ($H_2O_2$) and superoxide ($O_2\bullet-$), generating hydroxyl radicals (OH$\bullet$), which can damage various biological macromolecules (6). Additionally, prolonged oxidative stress can trigger the release of iron atoms from mononuclear enzymes, further increasing intracellular iron levels and exacerbating cellular damage (7). Therefore, bacteria must tightly regulate both iron homeostasis and oxidative balance to survive fluctuating environmental conditions.

To counteract oxidative stress, bacteria employ antioxidant defense systems composed of non-enzymatic and enzymatic components, along with the regulatory networks that govern them (1, 8). While enzymatic defenses include catalases and peroxidases targeting specific ROS, non-enzymatic defenses rely on small molecules with antioxidant properties (9), among which glutathione is the most well-characterized example (1, 10, 11).

Glutathione is the most abundant antioxidant molecule in cells and is present in virtually all bacteria and eukaryotic cells (4). It plays a crucial role not only in protecting against oxidative stress but also in maintaining cellular homeostasis, regulating sulfur transport, conjugating metabolites, detoxifying xenobiotics, conferring antibiotic resistance, regulating enzyme activity, and modulating the expression of stress response genes (11). At the molecular level, glutathione neutralizes ROS directly by donating electrons to $O_2\bullet-$, OH$\bullet$, peroxy radicals (ROO$\bullet$), and peroxynitrite (ONOO$-$), converting reduced glutathione (GSH) to glutathione disulfide (GSSG). Indirectly, glutathione also neutralizes ROS through enzymatic reactions involving glutathione peroxidase and glutathione reductase, which sustain its redox cycling capacity (4).

In bacterial pathogenesis, glutathione plays a pivotal role beyond its classical antioxidant function. In several pathogens, glutathione modulates virulence by activating transcriptional regulators, enhancing biofilm formation, and promoting resistance to oxidative stress (12–14). For instance, in *Listeria monocytogenes*, endogenous GSH acts as an allosteric activator of PrfA, the master virulence regulator (15), while in *Burkholderia pseudomallei*, host-derived GSH serves as a signal to trigger the expression of the Type VI Secretion System 5 during intracellular infection (16).

Despite these advances, further research is needed to clarify the molecular mechanisms by which glutathione shapes the global transcriptional and metabolic landscape of pathogenic bacteria. This question is particularly relevant in facultative anaerobes such as *Enterococcus faecalis*, a common gut inhabitant and opportunistic pathogen. *E. faecalis* has emerged as a well-established model for studying systems biology and heavy metal stress responses (17–21). Faced with an excess of iron, this bacterium activates or represses about 15% of its total genes, reflecting a global regulatory shift that includes genes related to both metal homeostasis and ROS defense.

In response to oxidative stress, *E. faecalis* activates sulfur metabolism and glutathione biosynthesis pathways, leading to elevated intracellular levels of GSH (22). In a previous study (21), it was demonstrated that iron exposure led to a marked decrease in total glutathione in *E. faecalis*, concurrently with the upregulation of antioxidant genes such as *sodA*, *katA*, and *msrA*. Moreover, several components involved in glutathione metabolism have been identified in this organism. Among them, glutathione reductase (GR), the enzyme responsible for catalyzing the reduction of GSSG back to GSH, has been biochemically and functionally characterized, providing insight into its critical role in counteracting oxygen-induced oxidative stress (23).

Although evidence supports the importance of glutathione in controlling oxidative stress in *E. faecalis*, there is still limited understanding of the global physiological consequences associated with glutathione, particularly under iron-induced stress conditions. To address this gap, we applied an integrative systems biology approach to investigate the global transcriptional and metabolic responses of the opportunistic pathogen *E. faecalis* under elevated iron levels in a glutathione-deficient scenario. To our knowledge, this is the first report to investigate the role of glutathione in the systemic response to iron exposure, providing a basis for future studies on its involvement in the pathogenesis of this bacterium.

## RESULTS

### Glutathione synthetase of *E. faecalis* is conserved in *Lactobacillales*

Typically, the synthesis of glutathione involves the sequential action of two enzymes: γ-glutamylcysteine synthetase (γGCS) and glutathione synthetase (GS) (24). However, in some species, such as *Streptococcus agalactiae*, a member of the *Lactobacillales* order, glutathione synthesis is catalyzed by a single enzyme that combines both functions: γ-glutamylcysteine synthetase–glutathione synthetase (γGCS-GS) (25, 26). In the *E. faecalis* genome, we identified a homologous γGCS-GS protein encoded by the monocistronic *gsh* gene (EF3089), which consists of an open reading frame of 2,271 nucleotides and encodes a 756-amino acid protein (Fig. 1A).

A comparative analysis of γGCS-GS across *Lactobacillales* species revealed a high degree of conservation in its primary structure (identity values > 45%) that is also present in other *E. faecalis* strains isolated from human patients (see Fig. S1 at https://github.com/Aliaga-Tobar/Supplementary_material.git). This suggests a common evolutionary origin shared with *Lactobacillus plantarum* and *S. agalactiae* (25), denoting the potential functional importance of γGCS-GS for *E. faecalis* in the context of human pathogenesis (12). In addition, to validate the predicted gene annotation *in silico*, we constructed a homology-based model using the crystallized structure of γGCS-GS from *Streptococcus agalactiae* (48% identity and 66% similarity between both protein sequences) (27). The resulting structural model (Fig. 1B) confirms a predominantly alpha-helical structure with approximately five beta-sheet domains, and the functional domain spanning residues 441–464, required for full γGCS activity, is well conserved in the *E. faecalis* enzyme. In addition, the substrate-binding region of γGCS-GS (residues 448–489), crucial for GS function in *S. agalactiae*, is also conserved in *E. faecalis*, reinforcing its predicted enzymatic role (25).

Regarding specific amino acids, we identified key residues involved in four processes relevant to enzymatic action: cysteine binding, glutamate binding, glycine binding, and magnesium cofactor binding. In *S. agalactiae*, the coordination of the magnesium cofactor involves three distinct sites, with residues Glu-29, Asp-60, Glu-67, His-150, Glu-328, and Glu-27. Additionally, the essential residue for glycine binding is an arginine located at position 588, the glutamate binding residues are Ile-146 (corresponding to residue 145 in *E. faecalis*), Arg-235 (not present), His-150, and the primary residues for cysteine binding in γGCS-GS—Phe-61, Tyr-131, and Leu-135—are also present, albeit with minor positional shifts (25, 28, 29).

In *E. faecalis*, Arg-588 (glycine binding), Tyr-131, and Leu-135 are located at Arg-592, Tyr-129, and Leu-133, respectively. Although these residues do not align perfectly with their counterparts in *S. agalactiae*, their conservation suggests they likely retain similar functional properties, warranting further experimental validation.

Overall, the strong sequence conservation within *Lactobacillales* and the presence of well-preserved functional residues support the conclusion that the annotated *gsh* gene in *E. faecalis* encodes a functional γGCS-GS enzyme, reinforcing its likely role in glutathione biosynthesis and oxidative stress response.

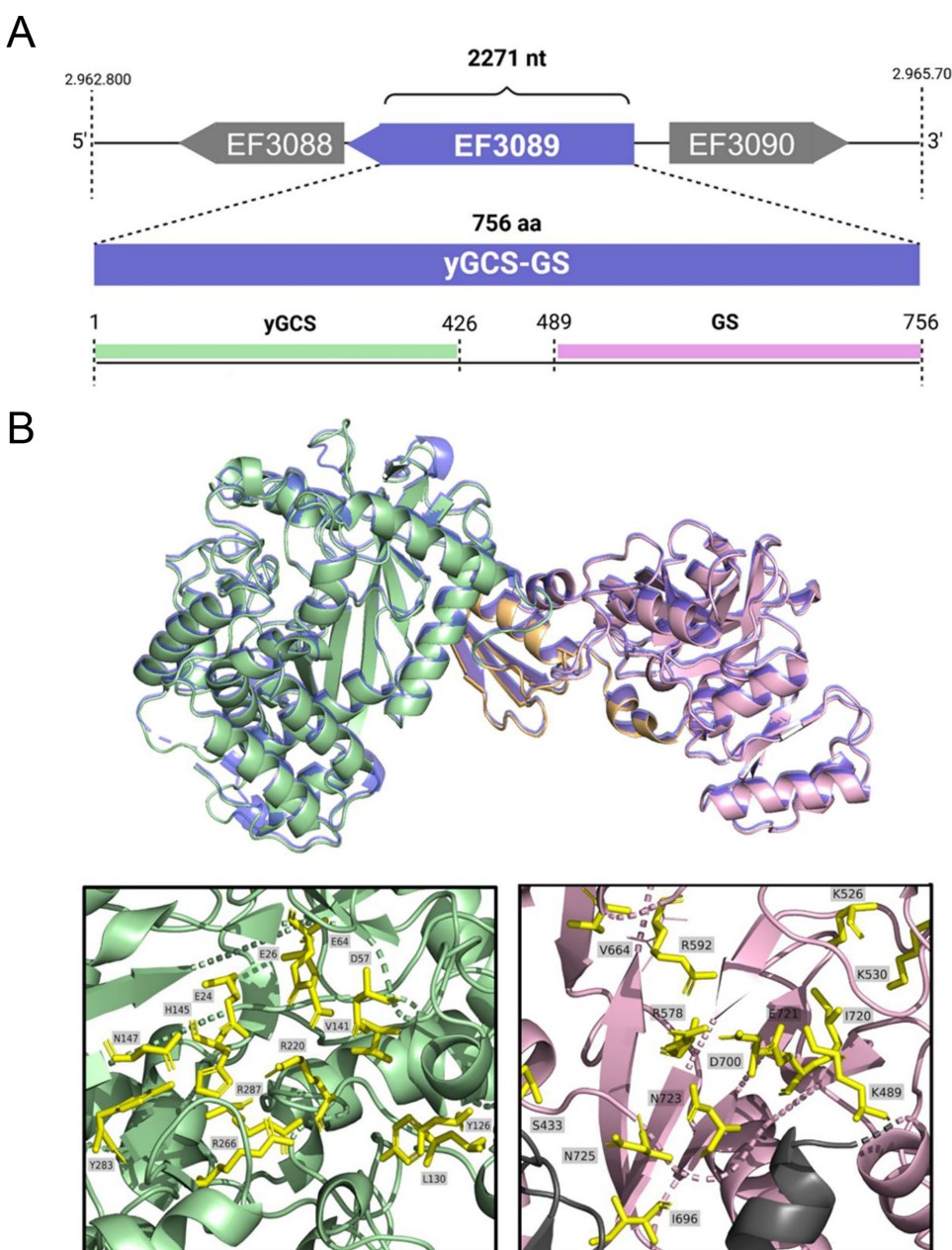

**FIG 1** Schematic representation of the bifunctional γGCS-GS enzyme from *E. faecalis*. (A) Genomic context of gene EF3089, encoding the bifunctional enzyme γGCS-GS. (B) Structural superposition of the γGCS-GS model from *E. faecalis* OG1RF (colored pale green for γGCS and light pink for GS) and the crystal structure of the homologous bifunctional enzyme from *S. agalactiae* (PDB ID: 3ln6.1, shown in blue), modeled using SWISS-MODEL. The interdomain region is highlighted in light orange. The lower panels show close-ups of the active sites of each domain, with conserved catalytic residues indicated in yellow and labeled accordingly.

## Absence of γGCS-GS in *E. faecalis* induces the reduction of intracellular glutathione and iron content

The increase in intracellular iron content induces the production of ROS through a series of chemical reactions (30, 31), but *E. faecalis* presents mechanisms to maintain homeostasis and responds to conditions of iron excess or deficiency (6). These mechanisms include transcriptional changes that promote the activation of genes and regulators related to basal metabolism, mainly controlled by Fur, a transcription factor responsible for regulating metal uptake systems to prevent excessive intracellular iron accumulation (6).

However, little is known about the mechanisms involved in the control of iron-generated toxicity in *E. faecalis*, a motivation that guides the goal of our study, which aimed to determine whether the absence of glutathione could be harmful to *E. faecalis* under conditions of iron excess.

To explore this, we studied the mutant strain for the γGCS-GS enzyme (Δ*gsh*) and compared its viability in relation to the wild-type (WT) strain under different levels of iron exposure. Surprisingly, the results (Fig. 2A) indicate no significant differences in viability between both strains, suggesting that the absence of the γGCS-GS enzyme does not impair growth, even under high iron exposure (4 mM, see Fig. S2 at https://github.com/Aliaga-Tobar/Supplementary_material.git). While the viability results did not show impaired growth in both Δ*gsh* and WT strains, given the importance of glutathione as an antioxidant, it is plausible to assume some compensatory effect that helps reduce iron toxicity.

In this context, turning our focus to the homeostatic response, we quantified the intracellular levels of glutathione (total, reduced, and oxidized) and iron in the WT and Δ*gsh* strains exposed to 0.5 mM FeCl$_3$. As anticipated, the Δ*gsh* strain exhibited a substantial 40% decrease in total glutathione content compared to the WT, primarily due to a twofold reduction in the levels of GSH (Fig. 2B; see Fig. S3 at https://github.com/Aliaga-Tobar/Supplementary_material.git). For its part, iron excess significantly diminished glutathione levels in the WT strain, reinforcing the idea that this molecule is actively consumed to counteract metal toxicity. In the Δ*gsh* strain exposed to Fe, glutathione levels showed an additional 30% reduction, which significantly impacts total GSSG content.

Although there are important changes in glutathione content due to the absence of the enzyme and the addition of Fe, these are not reflected in the viability of the bacteria under these conditions, which could imply the activation of complementary homeostatic pathways. To explore this compensatory response further, we measured the intracellular content of iron of each strain (Fig. 2C). Under basal growth conditions, no significant differences surfaced between WT and Δ*gsh* strains. However, under iron exposure, the Δ*gsh* strain exhibited a remarkable threefold reduction in intracellular iron levels compared to WT. This observation highlights a compensatory metal homeostatic strategy used by *E. faecalis*, whereby the bacterium reduces its iron content to cope with the reduced amount of glutathione produced in the absence of γGCS-GS.

In addition, despite the absence of the enzyme in the mutant strain, GSH could still be detected within the cell (both in the presence and absence of iron exposure). Figure 2B shows that the culture medium used for bacterial growth contains measurable concentrations of GSH. A possible explanation for this observation may lie in the presence of uptake mechanisms for this molecule from the extracellular environment. In *Streptococcus mutans* (order *Lactobacillales*), the protein GshT and the cystine ABC transporter complex TcyBC have been characterized as the main GSH importers (32). All these components are encoded in the *E. faecalis* genome, within the same operon (EF0804–EF0806; see Fig. S1 at https://github.com/Aliaga-Tobar/Supplementary_material.git). At the level of gene expression (Fig. 2D), there is a substantial increase in transcript abundance of all operon genes in the mutant strain exposed to iron. This result suggests, first, the active presence of possible components involved in the uptake of GSH from the extracellular medium, which would explain the levels of intracellular GSH in the mutant, and second, that these uptake systems may be activated in response to a critical GSH demand scenario. When intracellular GSH levels are extremely low, due to the combined effect of disrupted biosynthesis and GSH consumption for detoxifying excess iron, the activation threshold for GSH uptake systems may be reached, thereby sustaining the presence of intracellular GSH even in the absence of the γGCS-GS enzyme. Additionally, intracellular GSH content is also conditioned by its recovery from the GSSG molecule. In *E. faecalis*, the EF3270 gene has been described as encoding the GR enzyme (23, 33). In this regard, neither the absence of γGCS-GS in the mutant nor the addition of iron allowed us to detect changes in the transcript levels for this enzyme (Fig. 2D), suggesting

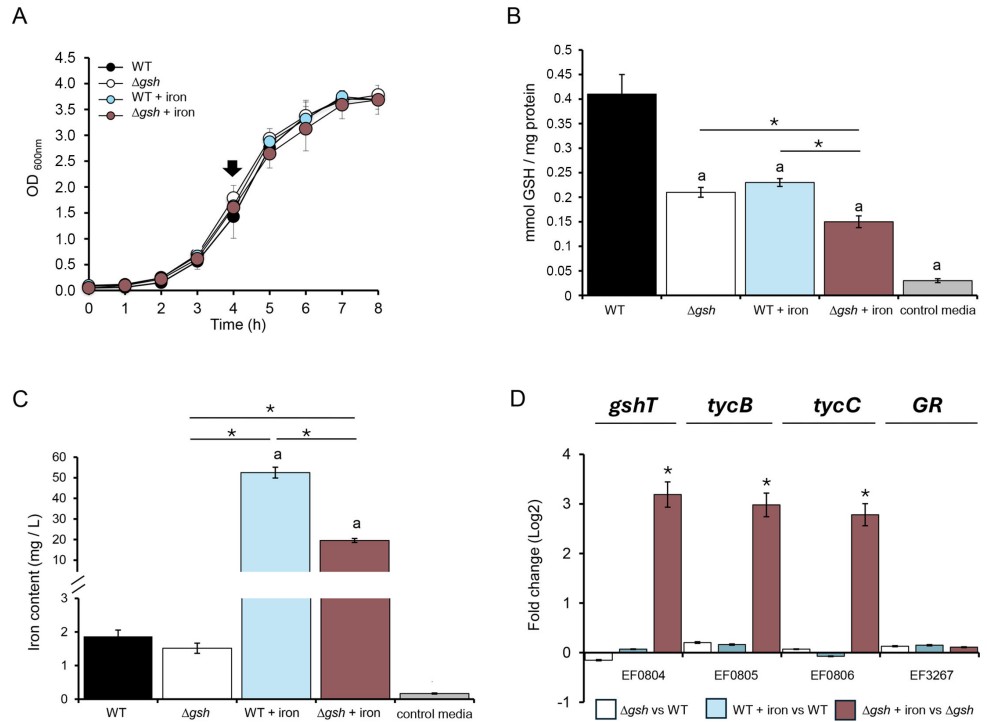

**FIG 2** Metal homeostatic response of *E. faecalis* WT and Δ*gsh* strains under iron excess. The iron exposure condition corresponds to 0.5 mM of FeCl$_3$ for 3 h. (A) Growth curves. The black arrow indicates the point selected for the quantification of glutathione and iron content. (B) Reduced glutathione quantification. (C) Intracellular iron content. Error bars represent standard deviation values. (D) Transcript changes of GSH uptake systems. Transcript levels were quantified by qPCR and are expressed as fold changes (Log2). In total, three biological replicates were performed (Mann–Whitney test, $P < 0.05$). Asterisk (*) denotes a significant difference. Letter a indicates significant differences against the WT strain growing in the control media.

that this process is not affected under the study conditions. This result further supports the view that the decrease in GSSG content is primarily due to the decrease in total GSH content resulting from the absence of the mutant and not to its recovery.

Considering the importance of glutathione and iron for the bacterium, our findings underscore a potential interplay between glutathione synthesis and iron homeostasis in *E. faecalis*, shedding light on global metabolic adjustments.

## *E. faecalis* specifically reconfigures its global transcriptional response in the absence of γGCS-GS

To determine whether the decrease in intracellular glutathione concentration affects the global transcriptional response, we performed a microarray gene expression assay. The analysis revealed a significant alteration in the transcriptional landscape when comparing the WT to the Δ*gsh* strain under basal growth conditions (Fig. 3A) with a total of 310 genes showing changes in expression in the absence of the γGCS-GS enzyme relative to WT. This strongly indicates that disruption of glutathione biosynthesis produces a broad transcriptional change compared to normal conditions with glutathione biosynthesis intact.

Under iron excess (3 h, 0.5 mM FeCl$_3$), the WT strain exhibited differential expression in 475 genes, reinforcing its well-documented sensitivity to iron availability fluctuations (6), although the most striking transcriptional reconfiguration was observed in Δ*gsh* strain under the same condition, where 483 genes exhibited altered expression patterns, reflecting a substantial shift that differs from the WT strain. While Fig. 3A illustrates a core subset of these differentially expressed genes (DEGs) based on stringent overlap criteria, only a small group of genes (*n* = 25) overlapped concordantly (i.e., up- or downregulated

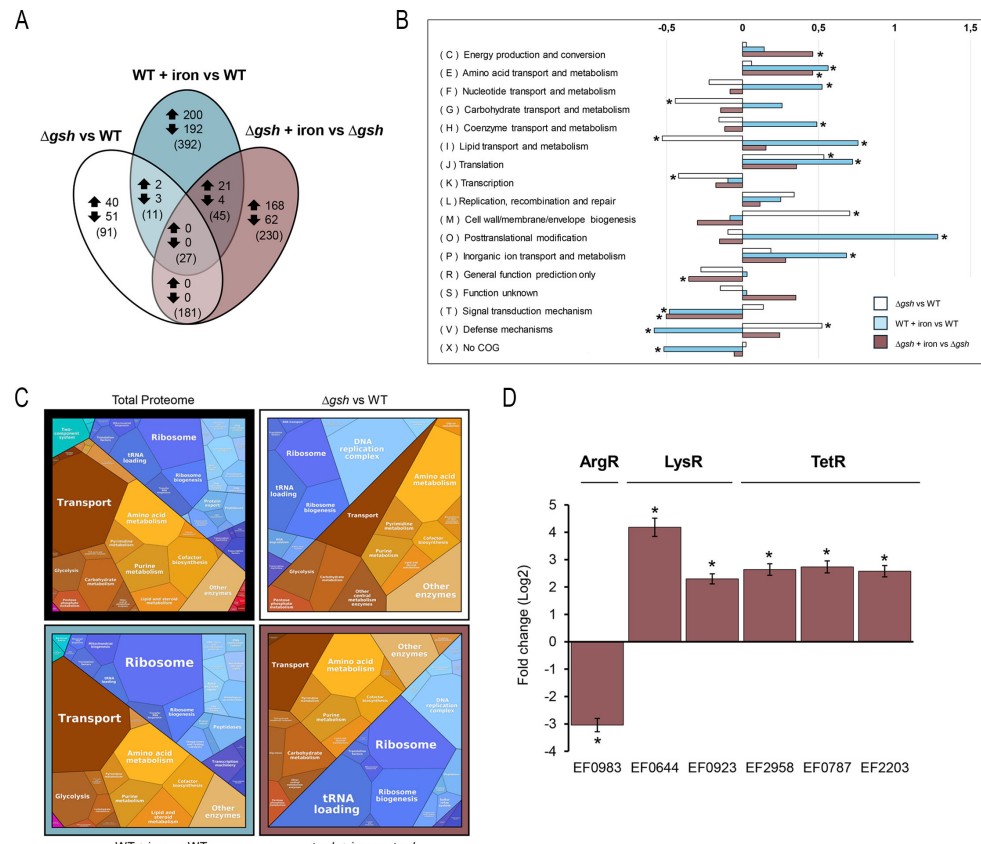

**FIG 3** Comparison of transcriptional changes of *E. faecalis* WT and *Δgsh* exposed to iron. (A) Venn diagram of sets of differentially expressed genes (DEGs). The diagram declares the number of genes that were upregulated (upward arrow) or downregulated (downward arrow) shared by the comparisons. Numbers in parentheses represent the total number of differentially expressed genes shared between the comparisons, including genes with concordant (arrows) and discordant regulation between conditions. (B) Enrichment analysis. The values are presented as Log2 of the total differentially expressed genes normalized by the total genes in the genome grouped per Clusters of Orthologous Groups (COG) category. Asterisk denotes significant enrichment (*P* < 0.05) compared to the total proportion in the bacterial genome. (C) Voronoi Treemaps. The construction of the maps was based on the complete proteome of *E. faecalis* OG1RF, using the list of DEGs identified for each experimental condition. KEGG reactions: total proteome: 758, *Δgsh*: 64, WT + iron: 165, and *Δgsh* + iron: 108. (D) Validation of mRNA abundance changes for transcription factors belonging to the ArgR, LysR, and TetR families. Transcript levels were quantified by qPCR and are expressed as fold changes (Log2) in the *Δgsh* mutant exposed to 0.5 mM FeCl$_3$ compared to the untreated *Δgsh* strain. Asterisks indicate statistically significant differences (Student's *t*-test, *P* < 0.05).

in both strains), highlighting the distinct regulatory adjustments driven by glutathione depletion. One possible explanation for this response is that the basal gene expression profile of both strains differs even before iron exposure, pre-conditioning the mutant strain to activate alternative regulatory mechanisms upon stress. In line with this, the reduction in the glutathione content appears to drive a global transcriptional shift that also correlates with decreased intracellular iron levels (Fig. 2C).

To gain further insights into the metabolic impact of glutathione depletion, we performed a Clusters of Orthologous Groups (COG) enrichment analysis and Voronoi Treemaps to examine the functions of overrepresented orthologous proteins and the KEGG pathways significantly enriched among the DEGs identified under all experimental conditions (Fig. 3B and C). As a result of the COG analysis, in the comparison between WT and *Δgsh* strains under basal conditions (white bars), we observed a decrease in the representation of differentially expressed genes linked to basal metabolism, including carbohydrates (*n* = 12) and lipids (*n* = 4), along with an increased representation of genes involved in membrane biogenesis and peptidoglycan synthesis (*n* = 18). In general

terms, after treatment with iron, the reduction in glutathione content generates a global metabolic change where processes linked to basal metabolism, such as nucleotide/carbohydrate biosynthesis and post-translational modifications, are significantly increased in both the WT and Δ*gsh* strains. Notably, the WT strain exhibited significant enrichment of DEGs in several basal metabolic categories, including carbohydrate metabolism, as well as post-translational modifications (light blue bars) that contrast with the Δ*gsh* strain that showed significant enrichment in the category related to energy production and conversion (burgundy bars). This suggests that Δ*gsh* strain may require additional ATP or reducing power to sustain antioxidant enzyme activity, highlighting a potential metabolic cost associated with compensating for glutathione deficiency.

Regarding the results of the Voronoi Treemap samples, the proteome coverage associated with the KEGG metabolic pathways that were affected at the transcriptional level in the WT strain exposed to Fe is 22%, which is expected given the total number of DEGs ($n = 475$) in proportion to the total *E. faecalis* genome ($n = 2,659$). However, the mutant with and without Fe exposure presents a considerable reduction in coverage, with 8% and 14%, respectively, of representativeness, although the total number of DEGs ($n = 310$ in Δ*gsh* and $n = 483$ in Δ*gsh* + Fe) maintains a similar proportion to the changes that occurred in the WT. This difference is reflected in the altered structure and size of the metabolic functions represented in the maps. In the WT strain exposed to iron, the organization and distribution of differentially expressed metabolic functions largely mirror those observed in the complete *E. faecalis* proteome, a structure that is markedly disrupted in the mutant strain, both under basal conditions and upon iron exposure. Notably, the mutant exhibits a significant increase in pathways related to amino acid and nucleic acid metabolism (including DNA replication and tRNA processing), along with a reduction in transport-associated functions. These changes may result from the depletion of glutathione in the mutant strain (a key antioxidant that participates in numerous metabolic processes), which could trigger compensatory shifts in alternative pathways, including some not yet categorized within KEGG. In this sense, these results suggest that glutathione depletion has a significant impact on the expression of genes involved in basal metabolism, implying transcriptional regulatory reconfiguration, particularly in the Δ*gsh* strain under iron stress. In terms of the transcriptional response of genes encoding transcription factors (Fig. 3D), in the Δ*gsh* strain exposed to iron, there is a decrease in the transcriptional abundance of the ArgR repressor (EF0983, regulator of arginine metabolism) (31, 34, 35), and an increase in the expression levels of global transcription factors belonging to the LysR families (EF0644, EF0923, and EF2958) and TetR (EF0787 and EF2203), which are involved in basal metabolism, energy generation, and general stress responses. This suggests that glutathione depletion induces regulatory adjustments at the transcriptional level, which could lead to important changes in the basal metabolism of *E. faecalis*.

## Low levels of glutathione impact amino acid concentrations

Considering the gene expression results, with the aim of identifying which metabolites could be affected by the decrease in glutathione levels, a metabolomic approach was carried out to determine the compounds involved in the response to iron exposure in both the WT and mutant strains (Table 1).

Overall, the decrease in glutathione levels impacts the basal metabolism of the bacteria with significant changes observed in nearly all quantified amino acids, as well as in NAD synthesis, nucleotide, and polyamine metabolism.

Under metal exposure, and consistent with the transcriptional level findings, significant changes were detected in the concentration of several key metabolites in the Δ*gsh* strain treated with iron, with an increase in the concentration of charged amino acids such as L-arginine and L-citrulline (metabolism of arginine) and polar amino acids such as L-serine. Notably, arginine metabolism can serve as a sole source of nitrogen, carbon, and energy (36) where the deimination of this amino acid enables the generation of ATP from ADP and phosphate while releasing ammonia (37). Interestingly, this

**TABLE 1**  Relative abundance of metabolites in *E. faecalis* WT and Δ*gsh* exposed to iron[a]

| Metabolite | Δ*gsh*/WT | WT + iron/WT | Δ*gsh* + iron/Δ*gsh* |
|---|---|---|---|
| L-serine | 0.71 ± 0.02 | 0.77 ± 0.04 | 1.2 ± 0.06 |
| L-aspartic acid | 0.8 ± 0.02 | 0.66 ± 0.02 | 0.82 ± 0.06 |
| Glycine | 0.67 ± 0.01 | 0.75 ± 0.01 | 0.87 ± 0.04 |
| Alanine | N.D. | N.D. | N.D. |
| L-glutamate | N.D. | N.D. | 0.73 ± 0.07 |
| L-citrulline | 0.6 ± 0.03 | N.D. | 1.22 ± 0.03 |
| Spermidine | 0.29 ± 0.07 | 0.33 ± 0.1 | N.D. |
| L-arginine | N.D. | N.D. | 1.24 ± 0.1 |
| L-histidine | 0.8 ± 0.05 | N.D. | 0.5 ± 0.07 |
| L-valine | 0.3 ± 0.13 | 0.73 ± 0.09 | 1.42 ± 0.8 |
| Methionine | N.D. | 1.14 ± 0.05 | 1.04 ± 0.06 |
| Hypoxanthine | 1.42 ± 0.02 | 1.52 ± 0.03 | 0.94 ± 0.02 |
| NAD | 2.05 ± 0.2 | N.D. | 0.34 ± 0.18 |
| L-isoleucine | 0.36 ± 0.06 | 0.64 ± 0.03 | N.D. |
| Inosine | 1.62 ± 0.06 | 1.4 ± 0.06 | 0.85 ± 0.01 |
| L-leucine | 0.59 ± 0.02 | 0.77 ± 0.03 | N.D. |
| Guanosine | 1.88 ± 0.02 | 1.73 ± 0.04 | 1.06 ± 0.01 |
| Adenosine | 1.93 ± 0.17 | N.D. | 0.63 ± 0.01 |
| L-phenylalanine | 0.73 ± 0.02 | 1.28 ± 0.04 | 1.37 ± 0.02 |
| Tyramine | 0.82 ± 0.01 | 0.68 ± 0.01 | 0.71 ± 0.01 |
| L-tryptophan | 0.84 ± 0.05 | N.D. | N.D. |
| Ribose | 1.3 ± 0.05 | 0.63 ± 0.09 | 0.59 ± 0.02 |

[a]The values represent the average of the rate of change between both conditions of three independent replicates. A *t*-test was used to evaluate statistical differences between means from the two conditions (N.D. indicates that there are no statistically significant changes between the samples compared, $P < 0.05$).

increase in arginine levels in the mutant exposed to iron directly correlates with the decrease in the abundance of the transcription factor ArgR (Fig. 3D), one of the major repressors of arginine metabolism genes.

The increase in arginine observed during the decrease in glutathione content could also be correlated with the detected increase in polyamines. Arginine and its precursor, ornithine, serve as substrates for synthesizing putrescine and spermidine, two major polyamines widely involved in ROS control, energy production, and protein synthesis (38–40). Conversely, there is a reduction in the concentration of metabolites related to ATP catabolism (inosine and hypoxanthine) and NAD in the mutant strain. The decrease in these compounds suggests greater utilization of these metabolites, which directly correlate with global gene expression assays, supporting the increased overrepresentation of genes involved in energy production and conversion mechanisms.

## Compensation pathways are related to arginine metabolism

To extend these findings at a systems level, we integrated our transcriptomic and metabolomic data into the *E. faecalis* genome-scale metabolic model (GSMM) published in 2015 (41), which was recently expanded and curated (42). The metabolic model of *E. faecalis* consists of 1,336 reactions and 1,293 metabolites, including all listed in Table 1. Regarding the impact of the absence of the γGCS-GS enzyme under basal conditions (see Fig. S4 at https://github.com/Aliaga-Tobar/Supplementary_material.git), there is an envelope expression of genes linked to purine and pyrimidine synthesis pathways, which directly correlate with the increase in metabolites associated with nucleic acids. This is coupled with the repression of genes associated with amino acid synthesis pathways, such as L-phenylalanine and L-tryptophan, explaining the decrease in these amino acids as a possible substitute for the decrease in intracellular glutathione content.

In response to metal treatment, Fig. 4A shows the integration of transcriptomics and metabolomics data within the metabolic model for both the WT and mutant strains

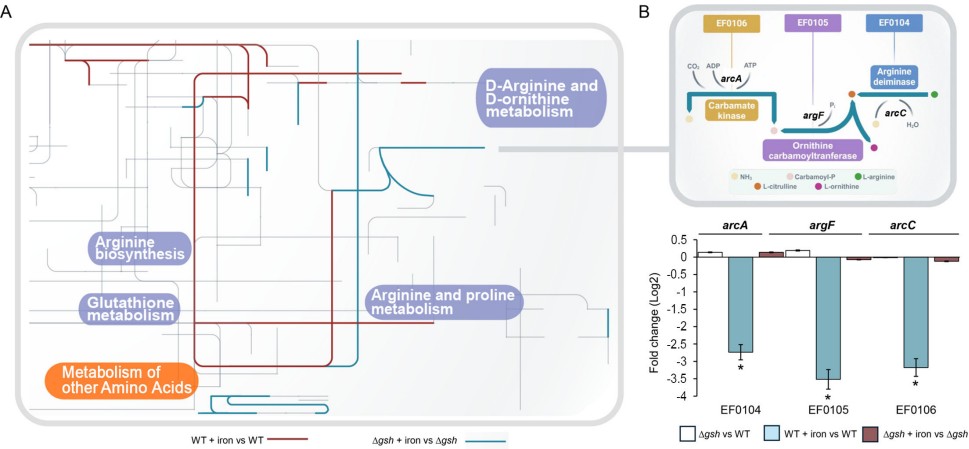

**FIG 4** Metabolic map of *E. faecalis* according to gene expression levels. (A) Pictorial representation of the fluxes involved in arginine metabolism in response to iron in the *E. faecalis* WT and Δ*gsh* (gray lines represent all *E. faecalis* metabolic pathways incorporated into the GSMM). Activated metabolic reaction in the model was marked based on the enzyme encoded by the differentially expressed gene from each condition (differentially expressed genes: light blue for WT + iron vs WT, burgundy for Δ*gsh* + iron vs Δ*gsh*). The integrated representation was generated using Ipath 3.0. (B) Metabolic pathway of the enzymes of the *arc* operon (arginine metabolism). Transcript levels of *arc* genes were quantified by qPCR and are expressed as fold changes (Log2). Asterisks indicate statistically significant differences (Student's *t*-test, $P < 0.05$).

exposed to iron (global view in Fig. S5 at https://github.com/Aliaga-Tobar/Supplementary_material.git). In both strains, metal exposure leads to the activation of metabolic pathways linked to arginine metabolism. In the WT strain, routes via the urea cycle lead to the generation of basal metabolites of the nucleotide type (pyrimidines) and amino acids such as alanine, aspartate, and glutamate. The latter are used in the synthesis of glutathione, in addition to the activation of routes associated with the basal metabolism of carbohydrates and fatty acids. In contrast, metabolism in the mutant strain derives through arginine synthesis pathways, which, as previously mentioned, seem to lead to the production of metabolites related to energy generation.

In this regard, in terms of the regulatory mechanisms and the expression of the systems associated with arginine metabolism, the increase observed in arginine levels (Table 1) in the mutant exposed to iron directly correlates with the decrease in the abundance of the repressor ArgR (Fig. 3C). This transcription factor could be regulating the *arc* operon (20), *arcA* (EF0104, encoding arginine deiminase), *argF* (EF0105, encoding ornithine carbamoyltransferase), and *arcC* (EF0106, encoding carbamate kinase), all of which are derepressed in the mutant compared to the WT (Fig. 4B).

According to these assumptions, arginine metabolism appears to act as a compensatory mechanism for reduced glutathione levels. To explore this, an *in silico* double mutant (Δ*gsh*Δ*arc*) was generated by removing genes related to arginine catabolism (*arc* operon; EF0104, EF0105, and EF0106) and glutathione synthesis (EF3089) from the metabolic model.

Using flux balance analysis (FBA) along with parsimonious criteria (pFBA), the optimal flux distributions for all model reactions and biomass production were calculated for the WT and double mutant models. Overall, there were no differences in biomass production between the two models (BM = 0.35 mmol/[gDW · h]), suggesting the robustness of this bacterium's metabolism to compensate for the elimination of specific metabolic pathways, including those associated with amino acid synthesis. Among the metabolic fluxes that were affected between the WT strain and the Δ*gsh*Δ*arc* mutant, the model predicted substantial alterations in glycerol metabolism (Fig. 5). The glycerol pathway is highly favored through an increase in the unidirectional flux of glycerol-3-phosphate synthesis, coupled with a decrease in the consumption of this metabolite, favoring the

generation of dihydroxyacetone phosphate, which is consumed through glycolysis to form pyruvate and subsequent energy production.

Overall, the simulated activation of both arginine and glycerol metabolism suggests the possible existence of compensatory mechanisms in *E. faecalis* that could help adjust bacterial metabolism to maintain cell viability under stress conditions such as iron exposure.

## DISCUSSION

The equilibrium between oxidants and reductants is crucial for the homeostasis of an organism. In this sense, iron, through the Fenton and Haber-Weiss reactions, amplifies oxygen toxicity by producing ROS (43). It is believed that tightly controlling iron metabolism, coupled with regulating defenses against oxidative stress, is essential in mitigating its harmful effects (30). In this context, *E. faecalis* has emerged as a prominent bacterial model for investigating transcriptional and metabolic adaptations in response to iron stress, offering valuable insights into the regulatory networks that coordinate metal homeostasis and oxidative stress responses (6, 21).

Here, we provide new evidence on how glutathione depletion reshapes the global transcriptional and metabolic responses of *E. faecalis* under iron excess. Prior research has established the essential role of glutathione in conferring tolerance to oxidative stress (10, 44, 45) and in mitigating the toxic effects of divalent metal ions (45). Our findings reveal that while glutathione deficiency does not impair bacterial viability, it triggers broad transcriptional and metabolic reprogramming, likely as a compensatory strategy.

### *E. faecalis* Δ*gsh* strain characterization under iron excess

According to our *in silico* and metabolic results and supporting previous analyses (25), *E. faecalis* utilizes a bifunctional γGCS-GS enzyme for glutathione biosynthesis, rather than the two-step pathway present in other bacteria and eukaryotes (46). This characteristic, also observed in *S. agalactiae* and *L. monocytogenes*, suggests an evolutionary adaptation that enhances glutathione synthesis efficiency (47). In addition, this enzyme is also conserved across multiple pathogenic *E. faecalis* strains isolated from human patients. This conservation suggests a potentially critical role of endogenous GSH in supporting *E. faecalis* survival and pathogenicity during infection, as described in other bacterial species (12, 48, 49). The deletion of γGCS-GS was not lethal for *E. faecalis*, consistent

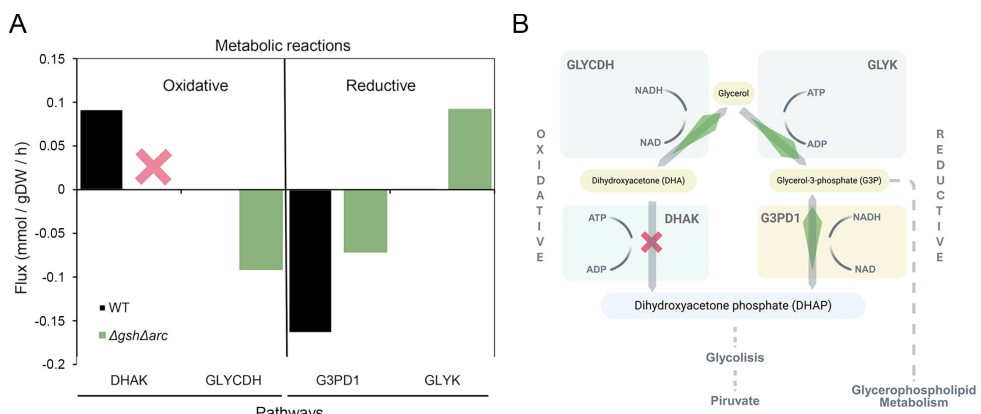

**FIG 5** FBA results and schematic overview of compensatory fluxes in the double mutant metabolic model. The effects of the double mutations were generated by removing the reactions directly from the metabolic model corresponding to each enzyme encoded by the genes *gsh* (γGCS-GS enzyme) and *arc* operon (anaerobic arginine catabolism). Reaction names: DHAK, dihydroxyacetone (glycerone) kinase; G3PD1, glycerol-3-phosphate dehydrogenase (NAD); GLYCDH, glycerol dehydrogenase; and GLYK, glycerol kinase. (A) Bars correspond to reaction fluxes of glycerol metabolism. (B) Schematic overview of glycerol metabolic fluxes. Increases in the thickness of the green arrows and the presence of red "X" marks indicate the directionality or blockage of reactions whose fluxes changed in the double mutant, respectively.

with observations in other bacteria such as *Pseudomonas aeruginosa* and *Streptococcus pneumoniae* mutants for glutathione biosynthesis (44, 45). In addition, the glutathione levels (GSH and GSSG) in the Δ*gsh* strain were significantly lower than in the WT, likely due to compensation through glutathione uptake and regeneration pathways (44, 45). However, these mechanisms appear less efficient than the *de novo* biosynthesis catalyzed by γGCS-GS; hence, the presence of intracellular glutathione is anticipated.

The decrease in the glutathione levels under iron exposure in both WT and Δ*gsh* strains is consistent with an oxidative stress response. Similar conditions (0.5 mM iron) have been shown to induce oxidative stress in *E. faecalis*, characterized by reduced glutathione levels and upregulation of antioxidant genes such as *sodA*, *katA*, *trx*, and *msrA* (21). Comparable oxidative stress responses have been observed in other bacteria, including *Acinetobacter baumannii* and *Escherichia coli* (7, 50).

In our case, we found comparable results during iron increase in the WT strain, showing a reduction in glutathione levels and upregulation of genes for thioredoxin reductase (EF1338) and peptide methionine sulfoxide reductase (EF1681) (see Fig. S6 at https://github.com/Aliaga-Tobar/Supplementary_material.git). Therefore, we suggest that *E. faecalis* faces a possible scenario of oxidative stress under iron exposure. In contrast, the Δ*gsh* strain with iron increase did not exhibit these expression changes observed in the WT, despite the reduction of glutathione levels. We attributed this to a compensation scenario that appears to be dedicated to avoiding an increase in intracellular iron levels, as indicated by the significantly reduced levels of intracellular iron in the Δ*gsh* strain compared to the WT.

In *P. aeruginosa*, glutathione mutants reduce iron uptake by decreasing siderophore production (44). Although the Δ*gsh* strain of *E. faecalis* did not show significant changes in iron uptake gene expression (6, 21), its reduced intracellular iron levels suggest compensatory mechanisms linked to protein efficiency rather than abundance. Additionally, glutathione may act as an intracellular iron buffer or retainer of Fe-S cofactors (51, 52). This could further support the observed phenotype in the mutant.

## Transcriptional reconfiguration of *E. faecalis* Δ*gsh* strain

Our analysis of the global transcriptional changes under glutathione depletion in *E. faecalis* revealed a broad transcriptional shift in the Δ*gsh* strain, and similar features have been reported in other microbial species. For example, a microarray analysis in *S. cerevisiae* showed that a mutant strain for γ-glutamylcysteine synthetase (GSH1), encoding the rate-limiting enzyme that participates in the first step of glutathione biosynthesis, presented differential expression of 189 genes under glutathione depletion (53). Similarly, a study using RNA-seq in the glutathione auxotrophic *Streptococcus pyogenes* (HKU16Δ*gshT* strain) revealed an extensive transcriptional perturbation under glutathione-decreasing conditions (11), showing expression changes in 165 genes, the majority of these genes (75%) downregulated, including several virulence-determinant genes. In this context, the different transcriptional response that we observed between *E. faecalis* WT and Δ*gsh* was expected.

Glutathione depletion induces a broad reprogramming of metabolic gene expression, as revealed by COG enrichment and Voronoi Treemaps. The Δ*gsh* strain shows reduced expression of genes related to basal metabolism and increased enrichment in energy production pathways, suggesting a metabolic burden linked to oxidative stress compensation. This shift may reflect an adaptive strategy to limit metabolic activity when glutathione is depleted, reducing ROS production (54). Similar responses have been observed in *A. baumannii* and glutathione-auxotrophic *S. pyogenes*, where metabolic downregulation occurs under oxidative stress (11, 50). Iron exposure amplifies these effects, especially in the mutant, which exhibits altered representation of functions like amino acid and nucleotide metabolism.

The transcriptional shift observed in *E. faecalis* implies the participation of transcriptional regulators. An example of this scenario was reported in *P. aeruginosa,* where the redox balance performed by glutathione upregulates T3SS gene expression via the

global transcription factor Vfr, which senses glutathione and activates T3SS expression (55). To our knowledge, no transcription factors have been reported to function in sensing glutathione levels in *E. faecalis*. In our analysis, however, the decrease in glutathione resulted in the downregulation of transcriptional repressor ArgR and the upregulation of transcription factors from the LysR and TetR families involved in basal metabolism, energy generation, and general stress. Moreover, a previous study introduced a global transcriptional regulatory model for *E. faecalis* (20), identifying that the promoter of the operon *arc* (EF0104, EF0105, and EF0106) contains a DNA binding site for the regulator ArgR, which is consistent with the transcriptional changes of these genes in the mutant under iron exposure. This suggests that transcriptional control mechanisms affected by glutathione levels are present in *E. faecalis*.

## Compensatory metabolic changes in *E. faecalis*

Our metabolic analysis revealed that under iron stress, the Δ*gsh* strain redirects its metabolic flux toward arginine synthesis pathways, apparently leading to energy generation and reductive power production. Thus, in a scenario where *E. faecalis* is faced with increased iron under glutathione deficiency, one way to mitigate the harmful effects of iron is to reduce intracellular levels of the metal. This metabolic shift may enhance energy production and support ROS detoxification.

Consistent with the transcriptional data, we observed a downregulation of *argR* in the Δ*gsh* strain under iron exposure, a known repressor of arginine catabolism genes, suggesting derepression of the arginine deiminase pathway. This transcriptional shift is paralleled by an accumulation of L-arginine in the mutant, as detected by metabolomic analysis. Together, these findings indicate that glutathione depletion leads to the release of ArgR-mediated repression, thereby activating arginine catabolism as a compensatory pathway to support energy production and redox balance under oxidative stress.

Unlike amino acids such as histidine, isoleucine, methionine, and tryptophan, which cannot be synthesized by *E. faecalis* (essential amino acids) (56), arginine synthesis pathways play a fundamental role in bacterial metabolism. According to our *in silico* predictions, the changes in metabolic fluxes in the mutant significantly impacted arginine synthesis and pathways related to NAD metabolism, indicating a redistribution toward stress protection in response to decreased glutathione content. This phenotype likely does not affect the overall biomass of the bacteria, as supported by viability assays, where no changes were observed in the growth of Δ*gsh* bacteria during metal exposure compared to the WT strain.

The derepression of the *arc* operon in the mutant strain under iron stress suggests that anaerobic arginine catabolism acts as a compensatory mechanism for glutathione deficiency. In this sense, arginine metabolism has been linked to oxidative stress resistance in several bacteria, including *Salmonella*, where it protects against ROS-induced damage (57, 58). Therefore, we hypothesize that L-arginine metabolism is part of the *E. faecalis* response to alleviate the ROS produced under conditions of increased iron and decreased glutathione.

Through the analysis of fluxes in the double mutant model, we propose that glycerol metabolism acts as a compensatory pathway in response to the loss of arginine biosynthesis, mediated by genes responsive to glutathione depletion. Glycerol catabolism in *E. faecalis* involves a dismutation process: blockage of the DHAK pathway leads to the accumulation of dihydroxyacetone, which is redirected toward glycerol via increased flux through the GLYCDH pathway. This reverse flux enables the reductive conversion of glycerol to 3-hydroxypropionaldehyde and subsequently to 1,3-propanediol, both steps showing increased flux in the double mutant. Despite the disruption of glutathione and arginine synthesis, biomass production remains unaffected, indicating robust metabolic compensation.

As a possible explanation, considering that *E. faecalis* is a facultative aerobic fermentative bacterium, its metabolism tends to ferment carbohydrates to produce energy, generating lactic acid as one of the final products through reductive reactions.

This fermentative tendency reinforces the importance of reductive routes under stress conditions. In this context, both the activation of the *arc* operon (linked to arginine fermentation) and the glycerol reductive pathway can be interpreted as parallel compensatory strategies to maintain redox balance and energy production in the double mutant.

*In silico* metabolic modeling represents a powerful approach to investigate physiological scenarios that are experimentally inaccessible, such as lethal gene deletions. The essentiality of the *arc* operon is supported by the lack of reported mutants for the *arcA*, *argF*, and *arcC* genes in other bacterial species, as well as by the complete absence of corresponding mutant strains for these genes in two independent genome-scale transposon mutant libraries of *E. faecalis* (59, 60). These observations highlight the value of computational simulations for exploring otherwise unattainable conditions and predicting metabolic adaptations under stress or genetic perturbations (61–64). In our case, the Δ*gsh*Δ*arc in silico* model allows us to explore metabolic adaptations that cannot be experimentally accessed due to the essentiality of the *arc* operon, providing insights into compensatory routes that may operate under partial inhibition or regulatory repression of arginine catabolism.

In summary, these results denote the putative existence of compensatory mechanisms in *E. faecalis*, which are directly involved in metal homeostasis (reduction of iron content) and others associated with transcriptional changes that directly correlate with increases or decreases in metabolic fluxes, primarily aimed at generating greater energy through the use of reductive pathways. All mechanisms aim to cope with the decrease in glutathione content in response to a redox-active agent such as iron. Notably, arginine metabolism emerges as a key adaptive strategy during these responses, facilitating survival under host-imposed stresses and potentially contributing to virulence traits such as biofilm formation and persistence within host tissues. Given the central role of GSH in maintaining redox balance and regulating iron homeostasis, future studies should explore how the interplay between GSH-dependent processes and arginine utilization shapes *E. faecalis* pathogenicity, which could guide the development of targeted therapeutic interventions aimed at disrupting these interconnected pathways to combat *E. faecalis* infections.

## MATERIALS AND METHODS

### *In silico* analysis of *E. faecalis* γGCS-GS protein

The γGCS-GS molecular 3D model was generated by the SWISS-MODEL program (65, 66), using information from the crystalline structure of γGCS-GS from *Streptococcus agalactiae* as a template. The final model was displayed with VMD version 1.9.4 software (67, 68). The γGCS-GS protein homologous in *Lactobacillales* order and *E. faecalis* organisms was recovered by BlastP on the National Center for Biotechnology Information website. The template gene was *E. faecalis gsh* gene EF3089. Global γGCS-GS alignments were performed using ClustalW as described in previous work (6).

### Strains, growth conditions, and iron treatments

All strains (*E. faecalis* OG1RF and Δ*gsh*) were grown in N medium (1% peptone, 0.5% yeast extract, 1% Na$_2$HPO$_4$, and 1% glucose) (65). Mutant glutathione strain (Δ*gsh*) of *E. faecalis* OG1RF was constructed using the pTEX4577 vector system (65, 67). Briefly, a 400-bp internal fragment of the *gsh* (EF3089) was amplified by PCR and cloned into the pGEM-T Easy vector (Promega). The insert was subsequently subcloned into the temperature-sensitive plasmid pTEX4577, generating the knockout construct. The construct was introduced into *E. faecalis* OG1RF by electroporation, and transformants were selected on N medium agar plates supplemented with 2 mg/mL kanamycin. Disruption of *gsh* was confirmed by PCR screening using primers flanking the gene locus. To assess potential off-target effects of the mutagenesis strategy on GSH levels, intracellular iron content,

and bacterial viability, we employed *E. faecalis* mutant strains for the CutC and LexA proteins (20, 65), previously generated using the same plasmid insertion approach (see Fig. S7 at https://github.com/Aliaga-Tobar/Supplementary_material.git).

For all the experiments, both strains were independently cultured overnight in N medium broth at 37°C and in triplicate (each replicate was conducted on different days). The next day, cells were diluted in two parallel cultures (control and iron) to a final $OD_{600}$ of 0.05 and then grown at 37°C and 160 rpm. The iron culture was supplemented with a concentration of 0.5 mM, as previously determined to be nonlethal (6), as well as 1, 2, and 4 mM of $FeCl_3$. Bacterial growth was monitored hourly over 8 h by measuring $OD_{600}$. For transcriptomic and metabolomic samples, samples were obtained from WT and *Δgsh* strains after 3 h with 0.5 mM of $FeCl_3$ and without iron added (control). Ferric chloride ($FeCl_3$) was selected as the iron source due to its previous use in oxidative stress models and its relevance in studies evaluating GSH-based protective mechanisms (69–71). Based on previous reports in *E. faecalis*, $FeCl_3$ concentrations above 0.5 mM do not lead to further significant increases in intracellular iron levels (21). Accordingly, a final concentration of 0.5 mM $FeCl_{33}$ was used in all assays, as it also falls within the physiological range of iron exposure encountered by *E. faecalis* in the human body (5, 72, 73).

## Intracellular iron and glutathione content

After the iron treatments (control or excess), the intracellular metal and glutathione content for each culture was determined as described below (74). Both strains, grown under different iron conditions and at the same growth stage (mid-exponential, 3 h of growth cultures according to the growth curves), were centrifuged and suspended in 200 µL of $HNO_3$ (Merck) and digested for 24 h at 65°C. After the acid lysis, total metal composition was determined by total reflection X-ray fluorescence. GSH and GSSG levels were quantified using the enzymatic recycling method described by Griffith (75), based on the reaction of GSH with Ellman's reagent (3-carboxy-4-nitrophenyl disulfide, DTNB) and its regeneration via GR in the presence of NADPH. After metal treatment, bacterial cultures (30 mL, in triplicate) were quenched and washed twice with cold glycerol-saline solution to induce lysis, followed by centrifugation at −20°C. Bacterial lysates were then prepared in phosphate-EDTA buffer, proteins were precipitated with 5% sulfosalicylic acid, and supernatants were used for analysis.

For total GSH/GSSG, samples were mixed with DTNB, NADPH, and GR, and absorbance at 412 nm was recorded (76). For GSSG determination, free GSH was derivatized with 2 µL of 2-vinylpyridine per 100 µL of sample and incubated for 45 min at room temperature. Quantifications were made in triplicate using calibration curves with GSH and GSSG standards. The molecules were then quantified as described above: GSSG was calculated by multiplying values by two, and reduced GSH was obtained by subtracting 2×GSSG from total glutathione. Quantifications were made in triplicate using calibration curves with GSH and GSSG standards. In addition, the content of GSH was quantified using the ortho-phthalaldehyde method (74), and the results are presented as nmol GSH/mg of total protein. Values represent the average data of three measurements for independent biological replicates. The statistical analysis was conducted using a Mann–Whitney test, $P < 0.05$.

## Global expression assays

Gene expression analysis was performed using a chip of four arrays of 72 K (catalog number A7980-00-01) from NimbleGen Systems, Inc. Each array contains 72,000 60-mer oligonucleotides, with an average of 11 oligos designed for the 3,114 open reading frames of *E. faecalis* OG1RF, representing twice the genome of the bacterium (two technical replicates per array). Next, for the WT and *Δgsh* strain, four independent hybridizations (two biological replicates per treatment paired with their respective controls) were performed by the manufacturer (Nimblegen) under equal conditions in a single glass, thus reducing variability between hybridizations (pre-hybridization,

hybridization, and washing steps). Scanning of the slides and data analysis were also carried out by Nimblegen. The gene expression from Fe-treated WT cells was compared to that of untreated cells; data from Fe-treated Δgsh mutant cells were compared to untreated Δgsh cells, and data from the Δgsh strain were compared to WT cells. Student's t-test statistics were used to identify significantly different gene expression levels with $P < 0.05$ and a fourfold magnitude of change between the average value of each gene and its corresponding reference, using the DNASTAR software Array star 3.0, similar to the previous report (65). Data available in the GEO database (ID GSE304768). For the gene enrichment analysis, the complete set of genes of *E. faecalis* OG1RF was used on BLASTP analysis against the COG database. Based on the retrieved annotations, an enrichment analysis of COG categories was carried out for the gene sets of *E. faecalis* OG1RF using Fisher's exact test with Benjamini–Hochberg multiple testing correction. Categories with corrected $P$-value <0.05 were considered enriched. Voronoi Treemaps were constructed using Proteomaps software (version 2.0) (77), based on Kyoto Encyclopedia of Genes and Genomes (KEGG) annotations for the DEGs obtained from transcriptomic data, a strategy previously applied in the analysis of the *E. faecalis* proteome (78). Detailed information on the genes, their identifiers, COG classification, KEGG, and differential expression values is given in Table S1 at https://github.com/Aliaga-Tobar/Supplementary_material.git.

For qPCR experiments, complementary DNA was synthesized from 2 µg of total RNA using Moloney Murine Leukemia Virus Reverse Transcriptase (Promega, USA) and random primers (Invitrogen). qPCR primers were designed with Primer3 Plus software (79), based on the *E. faecalis* OG1RF genome sequence. qPCR reactions and data analysis were performed as previously described (65), using the housekeeping gene *gdh* (EF1004) for normalization. Each reaction was conducted in triplicate with three independent biological RNA samples. Relative expression levels were calculated as fold changes and expressed as Log2. Statistical significance was evaluated using the Student's t-test or Mann–Whitney test ($P < 0.05$).

## Metabolomic data

The sampling, quenching, and intracellular metabolite extraction were done based on reported protocols (80). Briefly, 30 mL of culture broth was sampled in triplicate and quenched with cold glycerol-saline solution (3:2) and then centrifuged at −20°C. The cell pellets were resuspended in cold glycerol-saline solution (1:1) and centrifuged again at −20°C. Intracellular metabolites were extracted from the cell pellets using a cold methanol-water solution (1:1) at −20°C, followed by three freeze-thaw cycles. After centrifugation at −20°C, the supernatant was collected. The cell pellet was then resuspended in cold pure methanol (−20°C) for a second extraction, centrifuged at −20°C, and the supernatant was pooled with the previous one. The cell pellet was resuspended in bi-distilled water, centrifuged, and the supernatant was collected. Finally, 20 mL of cold bi-distilled water (4°C) was added to the metabolite extracts, which were then frozen and freeze-dried. Next, metabolite detection was realized by capillary electrophoresis coupled with electrospray ionization time-of-flight mass spectrometry. Normalization and quantification of metabolites were performed following previously validated analytic methods (60). Based on a reported analysis (81–83), to identify significant differences between mean values between conditions, a two-tailed t-test was performed assuming independence of groups and equal variances. The null hypothesis was rejected if the $P$-value was less than 0.05.

## *E. faecalis* GSMM analyses

For metabolic network construction, differentially expressed genes and selected metabolites were used to analyze the putative metabolic output through the GSMM available for *E. faecalis* (41, 42). Three *in silico* mutants were designed using this model. First, the glutathione mutant was generated by deleting the reactions associated with the gene EF3089 (*gsh*), corresponding to the reactions GLUCYSL and GTHS. For the

arginine mutant, the reactions linked to genes EF0104-0106, specifically *argA*, *argF,* and *argC* were removed. Finally, a double mutant was created by eliminating all reactions associated with both EF3089 and EF104-106 genes. For all these mutants, the maximum growth rate was defined as the objective function, and FBA, along with pFBA, was performed to calculate the optimal flux distributions across the metabolic network. Specific uptake rates were used as constraints, aligning the flux through the biomass equation with the specific growth rate (mmol gDW$^{-1}$ h$^{-1}$). This allowed FBA to determine optimal yield strategies, given that the maximum growth rate depends on the input fluxes (61, 84). Both FBA and pFBA were implemented using the COBRA Toolbox in MATLAB, with GUROBI employed to solve the optimization problems. iPath 3.0 was used to visualize the metabolic pathways of the GSMM (85). Pathways and specific reactions were highlighted based on changes in gene expression between the Δ*gsh* and WT strains, with and without iron exposure (transcriptomic data). Finally, the full code used for model construction and simulation, as well as the metabolic models used in this study, including those corresponding to the WT and mutant strains, is publicly available in the following GitHub repository: https://github.com/mathomics/E_faecalis_GSH_Iron.

## ACKNOWLEDGMENTS

This work was supported by Proyecto ANILLO regular ANID ACT210004 (V.A.-T., J.T., G.G., J.O., V.P., M.L.); Center for Mathematical Modeling, Apoyo a Centros de Excelencia ACE210010 (S.N.M., A.M., A.S., M.L.); ANID Millennium Science Initiative Program ICN2021_044 (V.A.-T., G.G., A.M., M.G., M.L.); Fondo Basal FB210005 (S.N.M., A.M., A.S., M.L.); ANID FONDECYT 1190742 (M.L.), 1230194 (V.P., M.L.), 1230195 (V.P., M.L.), and 1230724 (F.A.), Fondo Interdisciplinario y Proyecto Núcleo UOH CI2302 (G.G., J.O., J.T., M.L.); Convenio de Biodiversidad, UOH- Codelco DET (M.L.); FONDAP 1513001 (V.P.); Proyecto Postdoctoral ANID 3220080 (V.A.-T.) and UOH N° 3553/2025 (G.G); Beca Doctoral ANID 21220593 (J.O.), Universidad de Chile: grant Apoyo a la Infraestructura para la Investigación INFRA037/2023 (V.P.).

## AUTHOR AFFILIATIONS

[1]Centro de Genómica y Bioinformática, Facultad de Ciencias, Ingeniería y Tecnología, Universidad Mayor, Santiago, Chile

[2]Laboratorio de Bioingeniería, Instituto de Ciencias de la Ingeniería, Universidad de O'Higgins, Rancagua, Chile

[3]Centro de Biología de Sistemas para el Estudio de Comunidades Extremófilas de Relaves Mineros (SYSTEMIX), Universidad de O'Higgins, Rancagua, Chile

[4]Center for Mathematical Modeling, University of Chile, Santiago, Chile

[5]Millennium Institute Center for Genome Regulation, Santiago, Chile

[6]Laboratory of Cell Differentiation and Metabolism, Department of Biochemistry and Molecular Biology, Facultad de Ciencias Químicas y Farmacéuticas, Universidad de Chile, Santiago, Chile

[7]Advanced Center for Chronic Disease (ACCDiS), Facultad de Ciencias Químicas y Farmacéuticas and Facultad de Medicina, Universidad de Chile, Santiago, Chile

[8]Laboratorio de Microbiología Molecular, Departamento de Biología, Facultad de Química y Biología, Universidad de Santiago de Chile, Santiago, Chile

[9]Department of Mathematical Engineering, University of Chile, Santiago, Chile

[10]Equipe Dyliss, IRISA, Inria, CNRS, UMR 6074, Univ Rennes, Rennes, France

[11]Laboratorio de Bioinformática y Expresión Génica, INTA, Universidad de Chile, Macul, Santiago, Chile

## AUTHOR ORCIDs

Víctor Aliaga-Tobar http://orcid.org/0000-0002-4021-0112
Jorge Torres http://orcid.org/0009-0007-1431-419X

mSystems

Sebastián Nelson Mendoza 🔟 http://orcid.org/0000-0002-2192-5569
Jaime Ortega 🔟 http://orcid.org/0009-0008-0140-1355
Valentina Parra 🔟 http://orcid.org/0000-0002-0080-6472
Felipe Arenas 🔟 http://orcid.org/0000-0001-5604-5919
Alejandro Maass 🔟 http://orcid.org/0000-0002-7038-4527
Mauricio González 🔟 http://orcid.org/0000-0002-1592-9758
Mauricio Latorre 🔟 http://orcid.org/0000-0003-4746-8690

## FUNDING

| Funder | Grant(s) | Author(s) |
| --- | --- | --- |
| Agencia Nacional de Investigación y Desarrollo | ACT210004, ACE210010, ICN2021_044, FB210005,1190742, 1230194,1230195,1230724,1513001, 21220593, 3220080 | Víctor Aliaga-Tobar |
| | | Jorge Torres |
| | | Sebastián Nelson Mendoza |
| | | Gabriel Gálvez |
| | | Jaime Ortega |
| | | Sebastián Gómez |
| | | Valentina Parra |
| | | Felipe Arenas |
| | | Alejandro Maass |
| | | Anne Siegel |
| | | Mauricio González |
| | | Mauricio Latorre |
| Universidad de O'Higgins | Fondo Interdisciplinario, Proyecto Núcleo UOH CI2302, Convenio de Biodiversidad, UOH- Codelco DET, Proyecto Postdoctoral UOH N° 3553/2025) | Víctor Aliaga-Tobar |
| | | Jorge Torres |
| | | Sebastián Nelson Mendoza |
| | | Gabriel Gálvez |
| | | Jaime Ortega |
| | | Sebastián Gómez |
| | | Valentina Parra |
| | | Felipe Arenas |
| | | Alejandro Maass |
| | | Anne Siegel |
| | | Mauricio González |
| | | Mauricio Latorre |
| Universidad de Chile | INFRA037/2023 | Víctor Aliaga-Tobar |
| | | Jorge Torres |
| | | Sebastián Nelson Mendoza |
| | | Gabriel Gálvez |
| | | Jaime Ortega |
| | | Sebastián Gómez |
| | | Valentina Parra |
| | | Felipe Arenas |
| | | Alejandro Maass |
| | | Anne Siegel |
| | | Mauricio González |
| | | Mauricio Latorre |

## AUTHOR CONTRIBUTIONS

Víctor Aliaga-Tobar, Conceptualization, Data curation, Formal analysis, Investigation, Methodology, Validation, Writing – original draft, Writing – review and editing | Jorge Torres, Conceptualization, Data curation, Investigation, Methodology, Software, Visualization, Writing – review and editing | Sebastián Nelson Mendoza, Conceptualization, Data curation, Formal analysis, Methodology, Software, Visualization, Writing – review and editing | Gabriel Gálvez, Conceptualization, Data curation, Formal analysis, Methodology, Visualization, Writing – review and editing | Jaime Ortega, Conceptualization, Data curation, Formal analysis, Methodology, Visualization, Writing – review and editing | Sebastián Gómez, Conceptualization, Data curation, Formal analysis, Methodology, Visualization, Writing – review and editing | Valentina Parra, Conceptualization, Data curation, Methodology, Writing – review and editing | Felipe Arenas, Conceptualization, Data curation, Methodology, Writing – review and editing | Alejandro Maass, Conceptualization, Data curation, Methodology, Writing – review and editing | Anne Siegel, Conceptualization, Data curation, Writing – review and editing | Mauricio González, Conceptualization, Data curation, Writing – review and editing | Mauricio Latorre, Conceptualization, Data curation, Formal analysis, Funding acquisition, Investigation, Methodology, Project administration, Resources, Software, Supervision, Validation, Visualization, Writing – original draft, Writing – review and editing

## ADDITIONAL FILES

The following material is available online.

Open Peer Review

**PEER REVIEW HISTORY (review-history.pdf).** An accounting of the reviewer comments and feedback.

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
