## [Reviewer comments · mSystems]

Reduced glutathione levels in *Enterococcus faecalis* trigger metabolic and transcriptional compensatory adjustments during iron exposure.

Victor Aliaga-Tobar, Jorge Torres, Sebastian Mendoza, Gabriel Gálvez, Jaime Ortega, Sebastián Gómez, Valentina Parra, Felipe Arenas, Alejandro Maass, Anne Siegel, Mauricio González, and Mauricio Latorre

Corresponding Author(s): Mauricio Latorre, Universidad de OHiggins

Review Timeline:

Submission Date:	August 26, 2025
Editorial Decision:	October 17, 2025
Revision Received:	November 7, 2025
Accepted:	November 12, 2025

Editor: Shi Huang

Reviewer(s): Disclosure of reviewer identity is with reference to reviewer comments included in decision letter(s). The following individuals involved in review of your submission have agreed to reveal their identity: Rajib Saha (Reviewer #1); Jessica R Sheldon (Reviewer #2)

Transaction Report:

DOI: <https://doi.org/10.1128/msystems.01240-25>

Re: mSystems01240-25 (**Reduced glutathione levels in *Enterococcus faecalis* trigger metabolic and transcriptional compensatory adjustments during iron exposure.**)

Dear Dr. Mauricio A. Latorre:

Revision Guidelines

- Upload point-by-point responses to the issues raised by the reviewers in a file named "Response to Reviewers," NOT in your cover letter.
- Upload a compare copy of the manuscript (without figures) as a "Marked-Up Manuscript" file.
- Upload a clean .DOC/.DOCX version of the revised manuscript and remove the previous version.
- Each figure must be uploaded as a separate, editable, high-resolution file (TIFF or EPS preferred), and any multipanel figures must be assembled into one file.

Minireviews are not subject to publication charges.

Author Bios: We encourage you to submit a biographical sketch of each author (limit of 150 words) along with a photo to be published at the end of your article. You can submit these with your modified manuscript.

Figures Enhancement: ASM has engaged a professional science illustrator, Patrick Lane of ScEYence Studios, to work with minireview authors at the modification stage to generate improved figures that are uniform throughout the journal. This art enhancement service is free of charge to authors of minireviews and full-length reviews, and turnaround time is fast. I think you will be pleased with the results. Please contact Patrick on receiving this letter. Complete contact information for Patrick and further instructions are posted at <https://journals.asm.org/pb-assets/pdf-text-excel-files/graphical-enhancement-support.pdf>.

Sincerely,
Shi Huang
Editor
mSystems

Reviewer #1 (Comments for the Author):

The reviewer appreciates the authors for addressing the comments/concerns raised.

Reviewer #2 (Comments for the Author):

Firstly, the reviewer would like to thank the authors for their thorough, point-by-point account of how criticisms were addressed, and the sincere effort to speak to each concern. The manuscript has been considerably strengthened by the modifications made. Generally speaking, I am satisfied with the modified version of the manuscript, with the exceptions noted below.

General comments:

The results section is still somewhat choppy, with individual sentences standing alone as paragraphs. There are a number of places where sentences could be merged into complete paragraphs, for clarity. Similarly, there are a number of short paragraphs in the discussion that impede flow. Some examples are given below in the minor comments/corrections.

Glycerol metabolism seems to come out of nowhere in Figure 5. A better incorporation of the results and relevance of the GSMM could help with clarity. Further, if Δarc mutations are thought to be lethal, under what conditions would you expect to see the metabolic rearrangements predicted in the $\Delta gsh\Delta arc$ model?

Minor comments/corrections:

Lines 109-113. Firstly, this sentence is quite long and could be split into two. Secondly, it should be part of the previous paragraph because starting the paragraph with "At the molecular level, it neutralizes ROS..." does not refer back to anything to know what "it" is (presumably glutathione).

Line 111. Once the acronym for glutathione (GSH) is introduced, it should be used throughout the manuscript.

Line 120. T6SS5 is redundant with secretion system. Type VI secretion system 5 could be written out in full instead.

Lines 158-159. For clarity this could read "...encoded by the monocistronic gsh-like gene, EF3089." Although I am not entirely sure what is meant by "gsh-like," and *gsh* should be italicized.

Lines 162-167. Again, this is a rather long sentence, which could be divided into two. It would also make sense to join this single sentence with the paragraph below it.

Line 175. For clarity, make it obvious in this sentence that you are referring to γ GCS-GS. The sentence could be rephrased as "The substrate-binding region of γ GCS-GS (residues 448-489)..."

Line 183. It is unclear what is meant by "(there is in 145)."

Line 229. Change "measure" to "measured"

Line 275. Check the γ symbol before GCS-GS. It is showing as a square here, but nowhere else.

Line 307. Use "DEGs" instead of "differentially expressed genes" since the acronym has been introduced already.

Line 359. Add the word "and" before "polar amino acids"

Lines 379-382. This sentence needs to be reworked, as it stands it is somewhat confusing and the tenses don't match all the way through.

Line 430. Change "reaction" to "reactions."

Line 445. Remove italics from the delta symbol.

Lines 513-517. This paragraph should be joined to the one below it.

Lines 1135-1138. Check for tense agreement, as well as the use of singular vs. plural.

Line 1095. Change "gen" to "gene."

Line 1104. The statement "The iron exposure condition corresponds to 0.5 mM of FeCl₃ per 3 h" is confusing. Is iron added every 3 h?

Figure 2. Use decimals instead of commas in the number format on the y-axes.

Figure 3. It is unclear to me why the numbers in brackets where the Venn diagram has overlapping conditions do not add up to the numbers shown (e.g. in light blue, 2 genes are upregulated, and 3 genes are downregulated, but this adds to 11?). I might be missing something, but if these numbers are not supposed to add up - it would help to explain where they are coming from.

Figure 4. In the orange label, change "amino acid" to "amino acids"

Table 1. Is everything not marked "N.D." considered to be statistically significant? The caption for the table could be clearer. It is

unclear why red and green arrows are only shown in a handful of cases, even though there are larger changes seen.

Review of resubmission of Aliaga-Tobar *et al.* “Reduced glutathione levels in *Enterococcus faecalis* trigger metabolic and transcriptional compensatory adjustments during iron exposure”

Firstly, the reviewer would like to thank the authors for their thorough, point-by-point account of how criticisms were addressed, and the sincere effort to speak to each concern. The manuscript has been considerably strengthened by the modifications made. Generally speaking, I am satisfied with the modified version of the manuscript, with the exceptions noted below.

General comments:

The results section is still somewhat choppy, with individual sentences standing alone as paragraphs. There are a number of places where sentences could be merged into complete paragraphs, for clarity. Similarly, there are a number of short paragraphs in the discussion that impede flow. Some examples are given below in the minor comments/corrections.

Glycerol metabolism seems to come out of nowhere in Figure 5. A better incorporation of the results and relevance of the GSMM could help with clarity. Further, if Δarc mutations are thought to be lethal, under what conditions would you expect to see the metabolic rearrangements predicted in the $\Delta gsh\Delta arc$ model?

Minor comments/corrections:

Lines 109-113. Firstly, this sentence is quite long and could be split into two. Secondly, it should be part of the previous paragraph because starting the paragraph with “At the molecular level, it neutralizes ROS...” does not refer back to anything to know what “it” is (presumably glutathione).

Line 111. Once the acronym for glutathione (GSH) is introduced, it should be used throughout the manuscript.

Line 120. T6SS5 is redundant with secretion system. Type VI secretion system 5 could be written out in full instead.

Lines 158-159. For clarity this could read “...encoded by the monocistronic *gsh*-like gene, EF3089.” Although I am not entirely sure what is meant by “*gsh*-like,” and *gsh* should be italicized.

Lines 162-167. Again, this is a rather long sentence, which could be divided into two. It would also make sense to join this single sentence with the paragraph below it.

Line 175. For clarity, make it obvious in this sentence that you are referring to γ GCS-GS. The sentence could be rephrased as “The substrate-binding region of γ GCS-GS (residues 448-489)...”

Line 183. It is unclear what is meant by “(there is in 145).”

Line 229. Change “measure” to “measured”

Line 275. Check the γ symbol before GCS-GS. It is showing as a square here, but nowhere else.

Line 307. Use “DEGs” instead of “differentially expressed genes” since the acronym has been introduced already.

Line 359. Add the word “and” before “polar amino acids”

Lines 379-382. This sentence needs to be reworked, as it stands it is somewhat confusing and the tenses don't match all the way through.

Line 430. Change “reaction” to “reactions.”

Line 445. Remove italics from the delta symbol.

Lines 513-517. This paragraph should be joined to the one below it.

Lines 1135-1138. Check for tense agreement, as well as the use of singular vs. plural.

Line 1095. Change “gen” to “gene.”

Line 1104. The statement “The iron exposure condition corresponds to 0.5 mM of FeCl₃ per 3 h” is confusing. Is iron added every 3 h?

Figure 2. Use decimals instead of commas in the number format on the y-axes.

Figure 3. It is unclear to me why the numbers in brackets where the Venn diagram has overlapping conditions do not add up to the numbers shown (e.g. in light blue, 2 genes are upregulated, and 3 genes are downregulated, but this adds to 11?). I might be missing something, but if these numbers are not supposed to add up – it would help to explain where they are coming from.

Figure 4. In the orange label, change “amino acid” to “amino acids”

Table 1. Is everything not marked “N.D.” considered to be statistically significant? The caption for the table could be clearer. It is unclear why red and green arrows are only shown in a handful of cases, even though there are larger changes seen.

Dear Reviewer,

We would like to express our sincere gratitude for your second thoughtful and constructive evaluation of our manuscript entitled “*Reduced glutathione levels in Enterococcus faecalis trigger metabolic and transcriptional compensatory adjustments during iron exposure.*”

We truly appreciate the time and effort you devoted to reviewing our work, as well as the insightful comments that have helped us refine and strengthen the manuscript’s overall narrative and clarity.

All comments and suggestions have been carefully considered and thoroughly addressed in this revised version.

In particular:

The **Results and Discussion section was revised and modified according to comments.** The new version ensures a smoother narrative flow and improved grammar.

Ambiguous statements in Figures and the Table were clarified with additional context where necessary to improve reader comprehension.

Figures 2, 4 and 5 were updated based on comments.

Next, we provide the complete revision:

General comments:

The results section is still somewhat choppy, with individual sentences standing alone as paragraphs. There are a number of places where sentences could be merged into complete paragraphs, for clarity. Similarly, there are a number of short paragraphs in the discussion that impede flow. Some examples are given below in the minor comments/corrections.

R: We carefully modified these sections in order to make a narrative flow for the reader.

Glycerol metabolism seems to come out of nowhere in Figure 5. A better incorporation of the results and relevance of the GSMM could help with clarity. Further, if Δarc mutations are thought to be lethal, under what conditions would you expect to see the metabolic rearrangements predicted in the $\Delta gsh\Delta arc$ model?

R: In this final version, we have included a new sentence clarifying the inclusion of Glycerol metabolism (Lines 399-411). Regarding the reviewer’s question, the $\Delta gsh\Delta arc$ strain was modeled only as a hypothetical *in silico* perturbation, not as a viable mutant, to examine how *E. faecalis*’ metabolic network might adapt if arginine catabolism were severely limited during glutathione deficiency. Such constraint-based simulations are commonly used to study essential or synthetic-lethal deletions and identify compensatory pathways. (Orth *et al.*, 2010, DOI: 10.1038/nbt.1614; Thiele & Palsson, 2010, DOI: 10.1038/nprot.2009.203; Bordbar *et al.*, 2014, DOI: 10.1038/nrg3643; O’Brien *et al.*, 2015, DOI: 10.1016/j.cell.2015.05.019). This clarification has been added to the Discussion (Lines 561-572).

Minor comments/corrections:

We sincerely thank the reviewer for the suggestions and corrections. We carefully addressed each of these comments in detail, in particular the Result and Discussion section in which the text has been thoroughly reviewed accordingly to ensure that all suggestions have been fully incorporated to improve the quality of the manuscript.

1. Lines 109-113. Firstly, this sentence is quite long and could be split into two. Secondly, it should be part of the previous paragraph because starting the paragraph with "At the molecular level, it neutralizes ROS..." does not refer back to anything to know what "it" is (presumably glutathione).
R: The sentence was edited as suggested.
2. Line 111. Once the acronym for glutathione (GSH) is introduced, it should be used throughout the manuscript.
R: We regret this oversight. In the revised version, we have corrected this issue by using consistent terminology throughout the manuscript: "*glutathione*" to refer to total glutathione, "*GSH*" for reduced glutathione, and "*GSSG*" for oxidized glutathione.
3. Line 120. T6SS5 is redundant with secretion system. Type VI secretion system 5 could be written out in full instead.
R: The sentence was corrected as suggested.
4. Lines 158-159. For clarity this could read "...encoded by the monocistronic *gsh*-like gene, EF3089." Although I am not entirely sure what is meant by "*gsh*-like," and *gsh* should be italicized.
R: In this revised version, we have removed the suffix "-like" from the *gsh* name to avoid potential confusion.
5. Lines 162-167. Again, this is a rather long sentence, which could be divided into two. It would also make sense to join this single sentence with the paragraph below it.
R: The sentence was edited as suggested.
6. Line 175. For clarity, make it obvious in this sentence that you are referring to GCS-GS. The sentence could be rephrased as "The substrate-binding region of GCS-GS (residues 448-489)..."
R: The sentence was edited as suggested.
7. Line 183. It is unclear what is meant by "(there is in 145)."
R: The sentence was edited as suggested.
8. Line 229. Change "measure" to "measured"
R: Corrected.
9. Line 275. Check the symbol before GCS-GS. It is showing as a square here, but nowhere else.
R: Corrected.
10. Line 307. Use "DEGs" instead of "differentially expressed genes" since the acronym has been introduced already.
R: Corrected.
11. Line 359. Add the word "and" before "polar amino acids".
R: Corrected.
12. Lines 379-382. This sentence needs to be reworked, as it stands it is somewhat confusing and the tenses don't match all the way through.
R: The sentence was edited as suggested.

13. Line 430. Change "reaction" to "reactions."

R: Corrected.

14. Line 445. Remove italics from the delta symbol.

R: Corrected.

15. Lines 513-517. This paragraph should be joined to the one below it.

R: Corrected as suggested.

16. Lines 1135-1138. Check for tense agreement, as well as the use of singular vs. plural.

R: Corrected as suggested.

17. Line 1095. Change "gen" to "gene."

R: Corrected.

18. Line 1104. The statement "The iron exposure condition corresponds to 0.5 mM of FeCl₃ per 3 h" is confusing. Is iron added every 3 h?

R: Corrected as suggested.

19. Figure 2. Use decimals instead of commas in the number format on the y-axes.

R: Corrected for all Figures.

20. Figure 3. It is unclear to me why the numbers in brackets where the Venn diagram has overlapping conditions do not add up to the numbers shown (e.g. in light blue, 2 genes are upregulated, and 3 genes are downregulated, but this adds to 11?).

R: We have modified the figure legend to make explicit what the numbers in Figure 3A (Venn diagram) represent.

21. Figure 4. In the orange label, change "amino acid" to "amino acids"

R: Corrected.

22. Table 1. Is everything not marked "N.D." considered to be statistically significant? The caption for the table could be clearer. It is unclear why red and green arrows are only shown in a handful of cases, even though there are larger changes seen.

R: In this final version, the meaning of N.D. was clearly indicated. We fully agree with the reviewer's comment; the addition of arrows creates confusion and does not contribute to the discussion of the results; therefore they were removed from the Table.

We are confident that these revisions in the new version have significantly improved the manuscript, and we hope that it will now meet the standards for publication in mSystems.

Thank you again for your valuable feedback and consideration of our work.

Sincerely,

Victor Aliaga-Tobar
Assistant Professor
Center for Genomics and Bioinformatics, Universidad Mayor, Chile

Mauricio Latorre
Associate Professor
Director, SYSTEMIX Center
Universidad de O'Higgins, Chile

Santiago, 4th November, 2025

Re: mSystems01240-25R1 (**Reduced glutathione levels in *Enterococcus faecalis* trigger metabolic and transcriptional compensatory adjustments during iron exposure.**)

Dear Dr. Mauricio A. Latorre:

The work has been already of sufficient general interest and well written, given the importance of iron and oxidative stress in bacterial pathogenesis and substantial revisions.

Your manuscript has been accepted, and I am forwarding it to the ASM production staff for publication. Your paper will first be checked to make sure all elements meet the technical requirements. ASM staff will contact you if anything needs to be revised before copyediting and production can begin. Otherwise, you will be notified when your proofs are ready to be viewed.

Sincerely,
Shi Huang
Editor
mSystems